# Changes in Spo0A~P pulsing frequency control biofilm matrix deactivation

**Cristina S. D. Palma**[1], **Daniel J. Haller**[2], **Jeffrey J. Tabor**[1,2,3,4,5], **Oleg A. Igoshin**[1,3,5,6,7]*

1 Department of Bioengineering, Rice University, Houston, Texas, United States of America, 2 Systems, Synthetic, and Physical Biology Ph.D. Program, Rice University, Houston, Texas, United States of America, 3 Department of Biosciences, Rice University, Houston, Texas, United States of America, 4 Department of Chemical and Biomolecular Engineering, Rice University, Houston, Texas, United States of America, 5 Rice Synthetic Biology Institute, Rice University, Houston, Texas, United States of America, 6 Center for Theoretical Biological Physics, Rice University, Houston, Texas, United States of America, 7 Department of Chemistry, Rice University, Houston, Texas, United States of America

* igoshin@rice.edu

## Abstract

Under starvation conditions, *B. subtilis* survives by differentiating into one of two cell types: biofilm matrix-producing cells or sporulating cells. These two cell-differentiation pathways are activated by the same phosphorylated transcription factor - Spo0A~P. Despite sharing the activation mechanism, these cell fates are mutually exclusive at the single-cell level. This decision has been shown to be controlled by the effects of growth rate on gene dosage and protein dilution in the biofilm matrix production network. In this work, we explore an alternative mechanism of growth rate-mediated control of this cell fate decision. Namely, using deterministic and stochastic modeling, we investigate how the growth-rate-dependent pulsing dynamics of Spo0A~P affect biofilm matrix deactivation and activation. Specifically, we show that the Spo0A~P pulsing frequency tunes the biofilm matrix deactivation and activation probability. Interestingly, we found that DNA replication is the cell cycle stage that most substantially contributes to the deactivation of biofilm matrix production. Finally, we report that the deactivation of biofilm matrix production is not primarily regulated by the effects of growth rate on gene dosage and protein dilution. Instead, it is driven by changes in the pulsing period of Spo0A~P. In summary, our findings elucidate another mechanism governing biofilm deactivation during the late stages of starvation, thereby advancing our understanding of how bacterial networks interpret dynamic transcriptional regulatory signals to control stress-response pathways.

**Data availability statement:** A data software package with the deterministic and stochastic models used, and all figure generation scripts is deposited in GitHub (https://github.com/PalmaCristina/biofilm-matrix-deactivation).`

**Funding:** This work was supported by National Science Foundation [MCB-2204402 to O.A.I (co-PI) and J.J.T. (PI)], by the Jenny and Antti Wihuri Foundation [partially supporting postdoctoral stipend of C.S.D.P.] and by the National Science Foundation Graduate Research Fellowship [1842494 to D.J.H]. The funders had no role in study design, data collection and analysis, decision to publish, or preparation of the manuscript.

**Competing interests:** The authors have declared that no competing interests exist.

## Author summary

Bacteria have evolved various adaptation mechanisms to survive under challenging environmental conditions. For instance, under mild starvation, *B. subtilis* bacteria form biofilms —communities of cells encapsulated in a protective extracellular matrix. On the other hand, these bacteria differentiate into highly resistant spores under severe starvation. Interestingly, sporulating cells are often found within biofilm communities, but they do not contribute to biofilm matrix production. This is thought to be an energy-conservation strategy, as biofilm formation is an energy-intensive process and is therefore halted before sporulation begins. Though previous work has focused on the mechanisms driving biofilm disassembly, few studies have explored the regulatory processes that *B. subtilis* employs to halt matrix production before starting sporulation. In this study, we use mathematical models to demonstrate that the temporal dynamics of the biofilm master regulator Spo0A~P affect the deactivation of matrix production. Understanding the regulation of biofilm, a common lifestyle in bacteria, can lead to the development of synthetic strategies to either enhance or disrupt biofilm formation, with potential applications in medicine and industry.

## 1. Introduction

Cell fate refers to the specific developmental path that a living cell takes, leading to its ultimate differentiation into a particular cell type with specific functions. Transcription factors can drive cell fate decisions by activating or repressing genes related to alternative fates [1,2]. Understanding how transcription factors regulate cell-fate decisions is a fundamental question in biology because it determines how organisms maintain their functions and respond to the environment.

Bacteria, including the soil microbe *B. subtilis,* encounter challenging environmental conditions such as starvation. To ensure survival when facing starvation, *B. subtilis* can differentiate into multiple distinct cell types. In response to mild starvation, individual bacterial cells differentiate from a motile to a sessile cell state [3]. Cells in the sessile state produce and secrete an extracellular matrix, which leads to the formation of a biofilm [4]. Biofilms are characterized by matrix-encased communities of surface-associated cells that provide resistance to stresses such as antibiotics [4,5]. In contrast, prolonged starvation induces sporulation, wherein cells differentiate into metabolically-inactive spores that resist heat, chemical toxins, and other stressors [6–9].

The decision to form a biofilm or sporulate is governed by gene regulatory networks that are activated by the same major regulatory transcription factor, named Spo0A (0A) [10,11]. The concentration of 0A is controlled transcriptionally [12] and phosphorylation governs its activity post-translationally [2,13,14]. The phosphorylated form of 0A (0A~P) induces expression of sporulation and biofilm genes [2]. The phosphorylation of 0A is controlled by a phosphotransfer cascade (Fig 1A) termed

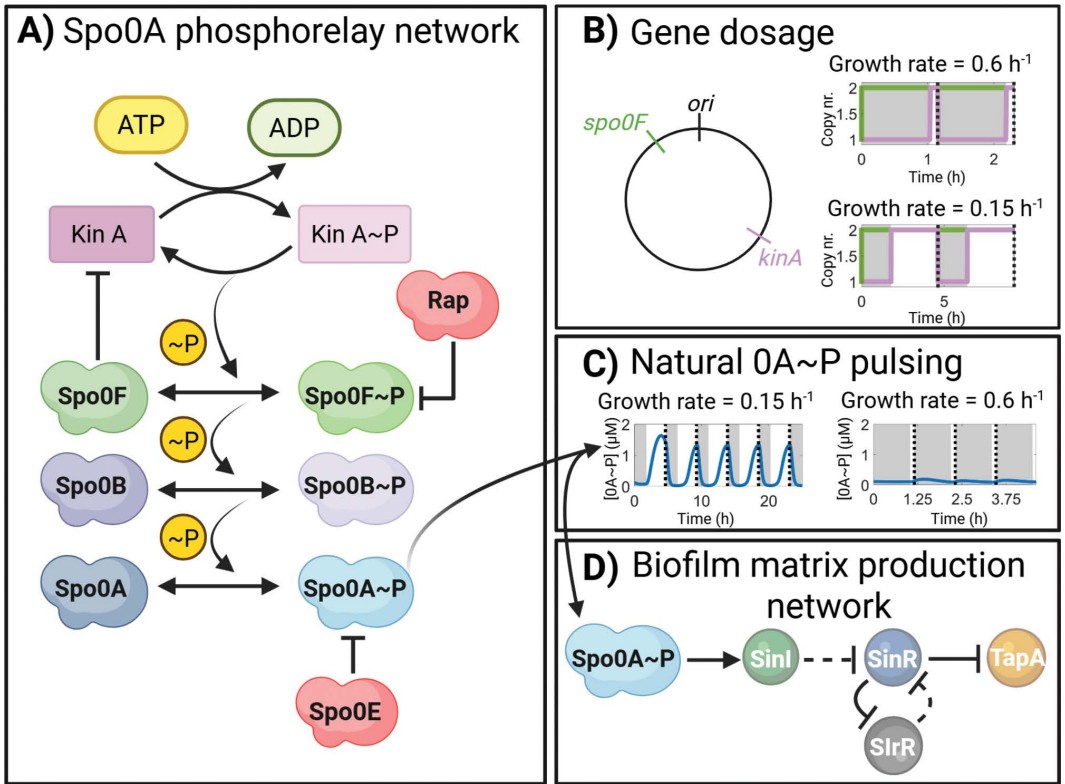

**Fig 1. Illustration depicting the networks analyzed. (A)** Post-translational phosphorelay network of Spo0A (0A) [14]. Kinase A (KinA) transfers phosphoryl groups to Spo0A via the two phosphotransferases Spo0B and Spo0F [18]. Phosphorylated Spo0F and Spo0A are subject to negative regulation by phosphatases Rap and Spo0E, respectively [19]. Spo0F inhibits KinA activity [20]. **(B)** Illustration of gene dosage imbalance for *kinA* and *spo0F*. Shown are the relative positions of *kinA* (126° - *terminus* proximal) and *spo0F* (326° - *oriC* proximal) in the genome [21]. Also shown are the approximate gene copy numbers over two cell cycles for a growth rate of 0.6 h⁻¹ (upper plot) and a growth rate of 0.15 h⁻¹ (bottom plot). Grey areas correspond to the DNA replication time, estimated using (Eq 9). Dashed lines correspond to the end of the cell cycle, estimated from (Eq 10). **(C)** Spo0A~P natural pulsing dynamics predicted by the model deterministic simulations of the phosphorelay network, for five complete cell cycles, assuming a growth rate equal to 0.15 h⁻¹. For comparison, also shown is the 0A~P predicted level, for four complete cell cycles, assuming a growth rate equal to 0.6 h⁻¹. **(D)** Biofilm matrix production network. Spo0A~P induces the production of SinI [22,23]. SinI represses the activity of SinR. SlrR and SinR form a double-negative mutual repressive network [24,25]. Dashed arrows depict regulatory effects due to post-translational interactions, i.e., the activity of SinR is regulated by complex formation with SinI and SlrR [24,25]. When SinR is in complex with SinI or SlrR, the expression of *tapA* and *slrR* is de-repressed. Figure created with BioRender [26].

phosphorelay [14]. The phosphorelay cascade initiates with five histidine kinases (KinA, KinB, KinC, KinD, and KinE) that autophosphorylate [15,16], with KinA being the major sporulation kinase [17]. Next, the phosphate from each of the kinases is transferred to two phosphotransferases, Spo0F (0F) and Spo0B (0B), and finally to 0A [14,18]. Further, 0A~P concentration is also controlled by the phosphatases Spo0E and Rap, which dephosphorylate 0A~P and 0F~P, respectively [19].

Previously, we demonstrated that 0A~P has pulsatile dynamics driven by a transient gene dosage imbalance and a negative feedback loop in the phosphorelay network [17]. Specifically, 0F inhibits KinA activity, and the chromosomal arrangement of *kinA* (terminus-proximal) and *spo0F* (oriC-proximal) (Fig 1B) leads to gene dosage imbalance during the DNA replication period. Since *spo0F* is replicated at the beginning of replication, while *kinA* is replicated near the end of the replication period, the DNA replication time is about the same length as the cell cycle during high growth rates [17], leading to a gene dosage ratio *kinA:0F* below 1:2 (upper plot in Fig 1B). These conditions lead to overall low levels of

0A~P in the cell (Fig 1C, right plot). Conversely, for low growth rates, the cell cycle is longer than the DNA replication time [17], leading to a balanced gene dosage ratio of 2:2, after DNA replication (bottom plot in Fig 1B). This leads to low 0A~P levels during the DNA replication period, followed by an overshoot after the replication period and prior to the cell division. In addition, the growth slowdown leads to an increase in the concentration of KinA, which further contributes to the overshoot of 0A~P amplitude [20]. This once-per-cell-cycle overshoot response leads to the pulsatile dynamics of 0A~P [17,20], as illustrated by the results of *in silico* simulations in Fig 1C (left plot).

Once phosphorylated, 0A can activate the matrix production genes via the biofilm matrix production network (Fig 1D), [11,27]. Namely, 0A~P induces the expression of transcription factor SinI, which is known to inactivate the transcription factor SinR by sequestration in a complex [22,23,28,29]. Active SinR inhibits the expression of SlrR, which can also inactivate SinR by sequestration [24,30]. Inactivation of SinR results in the derepression of a set of genes and operons controlling biofilm formation, including the *tapA* gene that encodes a protein required for biofilm matrix formation [24,31,32].

On the other hand, 0A~P is known to directly activate sporulation genes [33]. The activation of genes involved in sporulation requires a higher threshold of 0A~P than biofilm matrix production genes [2]. However, biofilm and sporulation as cellular fates are mutually exclusive (i.e., cells do not activate them simultaneously). Therefore, there should exist regulatory mechanisms that ensure biofilm matrix production deactivation when 0A~P levels exceed the threshold to trigger sporulation.

Recently, it was suggested that the biofilm matrix deactivation under starvation can be explained by the slowdown in growth rate [34]. In particular, the biofilm matrix production network was found to be bistable for 0A~P levels higher than a certain threshold. As growth slows down, there will be an insufficient amount of SinI and SlrR to fully inhibit the activity of SinR. As a result, the system enters a monostable region where matrix production is not active, i.e., *tapA* expression is low. However, that study focused on the population-level average concentration of 0A~P. Consequently, the impact of 0A~P pulsatile dynamics on the deactivation of biofilm matrix production remains unexplored at the single-cell level.

Here, we build on previous mathematical models [20,34] to predict how the biofilm matrix production network decodes 0A~P pulsing signals to regulate biofilm matrix production. Specifically, we examine how the frequency and amplitude of the pulsatile signal influence the biofilm matrix production network and, consequently, the cell-fate decision in *B. subtilis*.

## 2. Results

### 2.1 Biofilm is deactivated for a certain frequency and amplitude of the 0A~P natural pulsing signal

To investigate the effects 0A~P pulsing has on biofilm matrix deactivation, we coupled a deterministic phosphorelay network model (Methods Section 4.1, Tables A and B in S1 Text) [20] with a deterministic biofilm matrix production network model [34] (Methods Section 4.2, Tables C and D in S1 Text) and performed a deterministic simulation assuming decreasing growth rate (Fig 2A). We began by comparing the effects of 0A~P pulsing dynamics with those of a constant 0A~P level, corresponding to the average of the pulsing signal, across each cell cycle (Fig 2B). Starting from a matrix active state, our results show that for constant 0A~P levels, the production of the matrix protein TapA is not deactivated. However, assuming 0A~P pulsing, TapA production is deactivated earlier in time, i.e., at a faster growth rate (Fig 2C). In addition, we compared the effects of 0A~P pulsing dynamics to those of a constant 0A~P level set to either the maximum or minimum value of the pulsing signal over a cell cycle (Fig A in S1 Text). The results show that using the maximum value does not trigger matrix production deactivation. The minimum value falls below the threshold required for matrix activation, resulting in the deactivation of initially active cells. Overall, this suggests that the frequency and amplitude of 0A~P natural pulsing may act as a regulator of biofilm matrix deactivation. As such, not accounting for the pulsatile behavior of 0A~P may lead to potentially flawed conclusions about the mechanisms controlling cell fate decisions in *B. subtilis*.

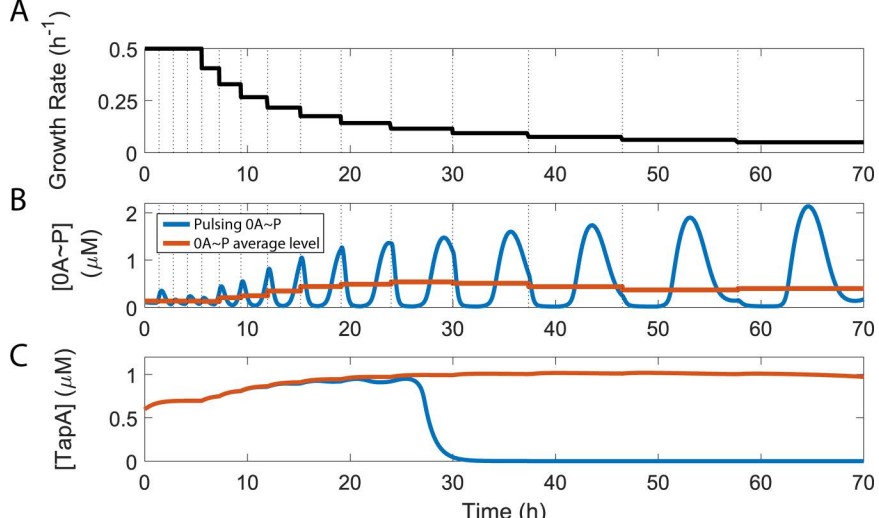

**Fig 2. Effects of 0A~P pulsing dynamics control biofilm matrix deactivation. (A)** Growth rate dynamics used as the input of the phosphorelay and biofilm matrix production mathematical models **(B)** 0A~P predicted levels (blue line) from the phosphorelay model, assuming the growth rate input dynamics shown in A. Also shown is the 0A~P pulse average level per cell cycle (orange line). Vertical dashed lines represent the beginning of a new cell cycle, according to Eq 10. **(C)** Predicted levels of biofilm matrix protein TapA, estimated from the biofilm matrix production network model. Blue line corresponds to the TapA predicted levels assuming the 0A~P pulsatile input shown in blue color in graph B. Orange line corresponds to the TapA predicted levels assuming the constant level of 0A~P, per cell cycle, shown by the orange line in graph B.

## 2.2 Pulsing dynamics of 0A~P affect the probability of matrix deactivation and activation

In single cells, matrix activation and deactivation are stochastic processes [34]. To examine how these processes depend on the 0A~P pulsing dynamics, we developed a stochastic model of the biofilm matrix production network, based on [34] (Methods Section 4.3, and Table E in S1 Text).

We performed stochastic simulations (Methods Section 4.4) using the natural pulsing 0A~P dynamics, assuming a growth rate equal to 0.4 h$^{-1}$ and the corresponding average signal (Fig 3A) as inputs. For each of the inputs, we simulated 2000 cells for a total of 40 hours, starting with cells in both a matrix-inactive state (*OFF*, Fig 3B) and a matrix-active state (*ON*, Fig 3C), at time = 0 h. We note that to model the cells' behavior accurately, each cell state at t = 0 h was set to be the final state of a 60 h preliminary stochastic simulation. The results show that, independently of the initial cell state, both input types can activate or deactivate the production of matrix protein TapA (Fig 3B and 3C).

Next, to investigate the effects of 0A~P pulsatile dynamics on matrix activation, we calculated the probability of matrix activation and deactivation over time. Namely, from the stochastic simulation results, at each time point, we calculated the fraction of cells with matrix active ($F^{ON}$):

$$F^{ON}(t) = \frac{N^{ON}(t)}{N^{OFF}(t) + N^{ON}(t)}$$

(Eq 1)

Here $N^{ON/OFF}(t)$ is the number of cells with matrix production active/inactive, respectively, at time $t$. We considered a TapA threshold ≥ 200 molecules to be associated with biofilm matrix production. This threshold was determined based on the bimodal distribution of TapA molecules (Fig B, left panel, in S1 Text). The fraction of active cells throughout the stochastic simulations is shown in light orange and light blue color in Fig 3D.

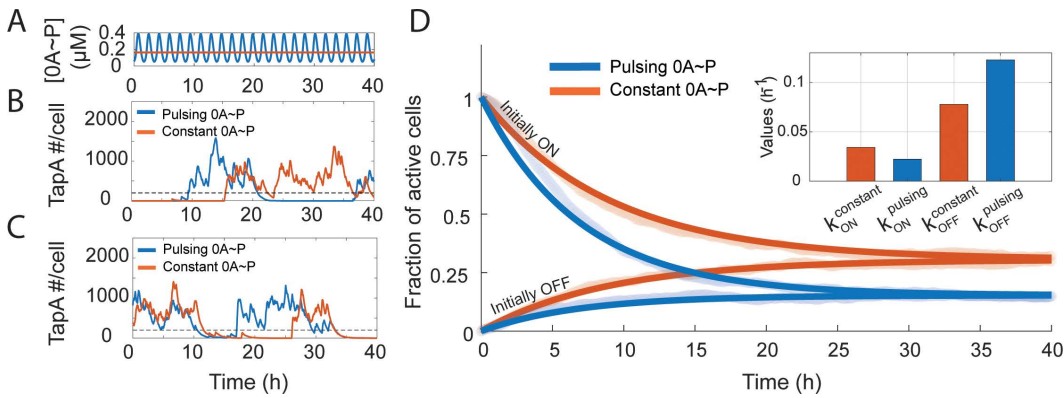

**Fig 3. 0A~P pulsing tunes the biofilm matrix deactivation rate. (A)** Pulsing (blue) and constant (orange) 0A~P signal used as input to the biofilm matrix production stochastic model. Pulsing signal corresponds to the signal for growth rate equal to 0.4 h⁻¹ predicted by the phosphorelay network model. The constant 0A~P signal corresponds to the mean of the pulsing signal. **(B)** Stochastic simulations starting from biofilm matrix inactive state for both constant and pulsing 0A~P signal. Matrix production is considered inactive if the number of TapA molecules < 200 (dashed line). **(C)** Stochastic simulations starting from matrix active state for both constant and pulsing 0A~P signal. Matrix production is considered active if the number of TapA molecules is ≥ 200 (dashed line). **(D)** Fraction of active cells as a function of time for constant (dark orange) and pulsing 0A~P signal (dark blue) estimated from fitting Eqs 3 and 4 to the stochastic simulation data of '*Initially OFF cells*' (light blue) and '*Initially ON*' (light orange) cells, respectively. All fits have $R^2 >= 0.90$ and MSE < 0.001. A total of 2000 simulations were performed. **(inset)** Estimated values of the biofilm matrix activation rate assuming constant ($k_{ON}^{constant}$) and pulsing 0A~P ($k_{ON}^{pulsing}$) and of the matrix deactivation rate assuming constant ($k_{OFF}^{constant}$) and pulsing 0A~P input ($k_{OFF}^{pulsing}$).

To understand the observed dynamics of $F^{ON}$ we employ a simple two-state model $ON \leftrightarrow OFF$ that assumes a Markov process (e.g., memory-less activation and deactivation) with effective rates $k_{ON}$ and $k_{OFF}$. For this model, we can write the probability of being in the active (*ON*) state as:

$$P_{ON}(t) = P_{ON}(0) \cdot e^{-t(k_{ON}+k_{OFF})} + \frac{k_{ON}}{k_{ON}+k_{OFF}} \cdot (1 - e^{-t(k_{ON}+k_{OFF})})$$

(Eq 2)

For $P_{ON}(0) = 0$ (*i.e.,* all cells have matrix production initially *OFF*), Eq 2 becomes:

$$P_{ON}(t) = \frac{k_{ON}}{k_{ON}+k_{OFF}} \cdot (1 - e^{-t(k_{ON}+k_{OFF})})$$

(Eq 3)

Meanwhile, for $P_{ON}(0) = 1$ (*i.e.,* all cells have matrix production initially *ON*), Eq 2 becomes:

$$P_{ON}(t) = e^{-t(k_{ON}+k_{OFF})} + \frac{k_{ON}}{k_{ON}+k_{OFF}} \cdot (1 - e^{-t(k_{ON}+k_{OFF})})$$

(Eq 4)

At steady state (*i.e.,* as $t \to \infty$) both Eq 3 and Eq 4 approach the same steady state fraction of active cells:

$$P_{ON}(\infty) = \frac{k_{ON}}{k_{ON}+k_{OFF}}$$

(Eq 5)

To estimate $k_{ON}$ and $k_{OFF}$ rates based on the whole range of simulated times, we simultaneously fit Eq 3 to '*Initially OFF*' cells and Eq 4 to '*Initially ON*' cells (Methods Section 4.5). Fitting results (dark orange and blue lines in Fig 3D) yield a good fitting (MSE < 0.001), for both pulsing and constant 0A~P input. The fitting suggests that, for pulsing 0A~P, $k_{ON}$ is

approximately ~0.01 h⁻¹ lower than for the constant 0A~P signal. Meanwhile, $k_{OFF}$ shows an absolute increase of ~0.07 h⁻¹ for the pulsing 0A~P signal compared to the constant signal. However, the relative change in both rates is approximately 50%. As such, we conclude that 0A~P pulsing dynamics contribute to biofilm matrix deactivation by increasing the deactivation rate ($k_{OFF}$) and decreasing the activation rate ($k_{ON}$). To test the robustness of the estimations, we tested whether the estimated $k_{ON}$ and $k_{OFF}$ rate constants are affected by the choice of threshold value. To this end, we performed 100 analyses using randomly selected thresholds ranging from 25% to 175% of the original 200-molecule value. The resulting mean and standard deviations of $k_{ON}$ and $k_{OFF}$ suggest that these estimates are not substantially affected by the chosen threshold (Fig B, right panel, in S1 Text). In addition, we tested whether additional sources of noise, such as the accumulation of positive DNA supercoiling [35,36], significantly affect the results. To this end, we extended the model by adding three reactions that allow the promoter to enter a 'locked' state in which the transcription factor cannot bind, and transcription cannot occur (Table F in S1 Text). We performed 500 stochastic simulations per condition (Methods Section 4.4). The results are robust, supporting the conclusion that 0A~P pulsing dynamics contribute to biofilm matrix deactivation (Fig C in S1 Text). Moreover, we also performed simulations assuming a stochastic model that includes stochasticity in cell cycle dynamics (Methods Section 4.4). Overall, we found that incorporating cell cycle stochasticity does not alter the effects of 0A~P pulsing in biofilm matrix deactivation (Fig D in S1 Text).

## 2.3 Period of 0A~P oscillation tunes the biofilm matrix deactivation threshold

Reference [34] investigated the bistability of the biofilm matrix production network, without taking oscillations of 0A~P into account. Namely, for a given concentration of 0A~P, a deterministic model of the network predicts the existence of two stable steady states of TapA levels. For a 0A~P concentration lower than ~0.1 µM the system is monostable with only a matrix *OFF* steady state (Fig 4A). For a 0A~P concentration higher than ~0.1 µM the system is bistable, with an *OFF* and an *ON* matrix stable steady-state of TapA (Fig 4A, grey area).

To examine the effects of 0A~P pulsing dynamics on the biofilm deactivation threshold (i.e., the average 0A~P concentration for which the system shifts from a bistable to a monostable *OFF* state), represented by the dashed line in Fig 4A, we generated *in silico* 0A~P oscillatory signals with varying amplitudes and periods. This approach allowed us to investigate how these parameters influence the deactivation threshold by decoupling the effects of amplitude and period from the natural pulsing behavior of 0A~P. In Fig 4B, we show the effects that three different 0A~P oscillatory input signals have on the deactivation threshold. Specifically, periods ($T$) of 1, 3, and 6 hours were tested. The results show that as the period increases (*i.e.*, the frequency decreases), the matrix deactivation threshold also increases. This implies that a longer oscillation period requires a higher amplitude of the 0A~P signal to maintain the matrix production (*ON*) state. Notably, the threshold is almost unresponsive to changes in period for short oscillation periods but becomes highly responsive to longer periods (Fig 4C). This indicates that the system deactivation threshold exhibits varying degrees of responsiveness depending on the oscillation period. Altogether, these results suggest that as the period increases, the amplitude of the 0A~P oscillatory signal must increase significantly to sustain biofilm matrix production.

Next, we examined how the deactivation threshold is affected when there is one 0A~P pulse per cell cycle, *i.e.,* the situation reflecting natural conditions [17]. For this study, we began by analyzing how the deactivation threshold varies with the cell cycle length for oscillatory periods of 1, 3, 5, and 6 hours (light blue lines in Fig 4D). Next, for each line, we selected the value for which the oscillation period matches the cell cycle length and fit a power-law approximation (Methods Section 4.5). The best fit line (dark blue line in Fig 4D) shows that the biofilm deactivation threshold increases as a function of the cell cycle length if one pulse occurs per cell cycle.

Finally, to examine the effects of pulsing versus a 0A~P constant signal on matrix deactivation across different growth rates, we compare the deactivation threshold as a function of the growth rate for an oscillatory signal occurring once per cell cycle (estimated from the best-fit line in Fig 4D) with that of a constant 0A~P signal. For both types of signals, the results (Fig 4E) show that as cells grow slower, the required level of 0A~P to keep the matrix active (*ON*) becomes higher.

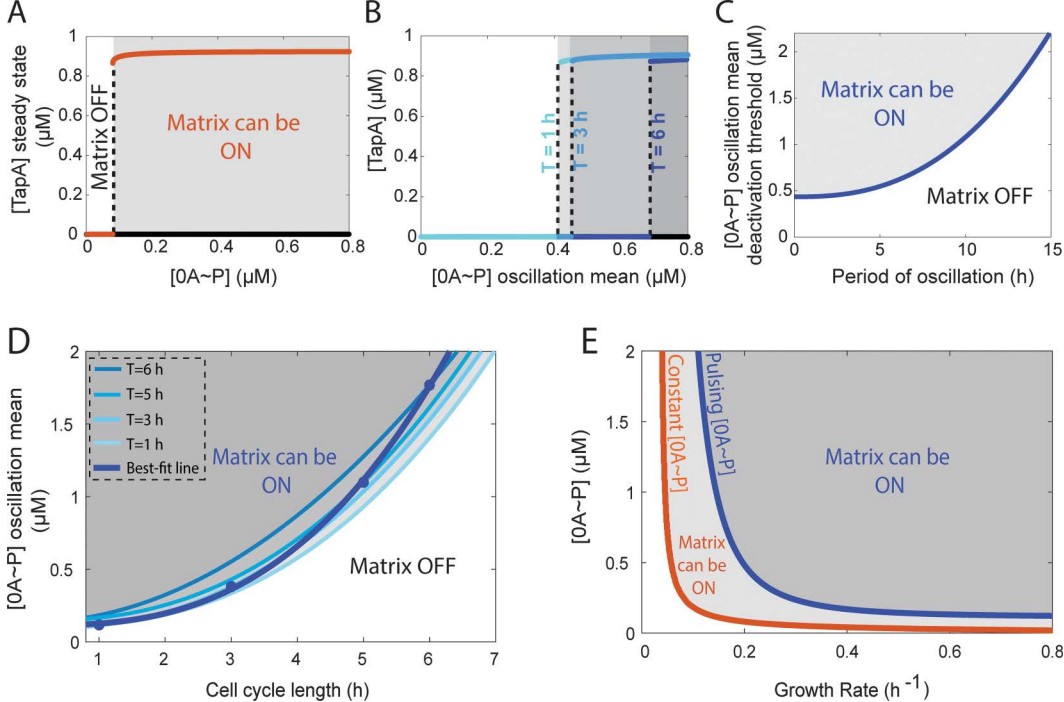

**Fig 4. Biofilm matrix deactivation threshold increases with increasing 0A~P oscillation period. (A)** Bistability diagram of the biofilm matrix production network assuming a constant level of 0A~P. The dashed line represents the concentration of 0A~P that corresponds to the biofilm matrix deactivation threshold. To the right of the dashed line the system presents bistability where two steady states co-exist: one corresponding to high levels of matrix protein TapA and the other corresponding to deactivated biofilm matrix production (i.e., low levels of TapA). For 0A~P concentrations below the threshold the system is monostable with low levels of TapA production (matrix production *OFF*). The results shown are for a growth rate of 0.2 h⁻¹. **(B)** Biofilm deactivation threshold as a function of the 0A~P oscillation mean for three oscillatory signals of different periods. Oscillations were programmed as cosine functions of the form $[0A \sim P] = amplitude \cdot \cos(freq \cdot t) + offset$. The oscillation periods tested were 1, 3, and 6 h, which correspond to an angular frequency (*freq*) of ~6.28 h⁻¹, 2.09 h⁻¹, and 1.05 h⁻¹, respectively. The *offset* was set such that the minimum value of the [0A~P] oscillation is 0. The results shown are for a growth rate of 0.2 h⁻¹. **(C)** [0A~P] biofilm deactivation threshold (i.e., the 0A~P oscillation mean amplitude) as a function of the period of oscillation. **(D)** Bifurcation between regions for which the matrix can be active (*ON*) and inactive (*OFF*) as a function of the cell cycle length, assuming 0A~P oscillatory signals of different periods (*T*). Shown is the bifurcation deactivation threshold for *T*=1, 3, 5, and 6 h (light blue lines). Also shown is the best-fit power-law approximation (Methods Section 4.5) estimated from the datapoints for which the oscillation period matches the cell cycle length (dark blue line). **(E)** Bifurcation between regions for which matrix can be active (*ON*) and inactive (*OFF*) as a function of growth rate. The orange line shows the result assuming a constant level of 0A~P. The blue line shows the result assuming a period of oscillation of 0A~P equal to the cell cycle length (estimated from the dark blue line in D).

However, the effects of oscillatory dynamics significantly affect when the matrix production is deactivated, causing it to occur earlier (i.e., higher growth rates) compared to constant 0A~P dynamics.

## 2.4 DNA replication time increase is a key determinant factor for biofilm deactivation

Given that the natural 0A~P pulsing period matches the cell cycle duration (Fig 1B) and matrix deactivation occurs at a certain cell cycle length, we investigated which phase of the cell cycle (DNA replication or post-replication) contributes most to matrix deactivation.

We started by building a simple model where the 0A~P input is set to be a square pulsing signal. We tested two different square pulsing signals. For the first signal, the 0A~P *ON* time is kept constant, and the *OFF* time increases in each cycle (Fig E, upper left panel, in S1 Text). For the second signal, the opposite occurs (Fig E, bottom left panel, in S1 Text).

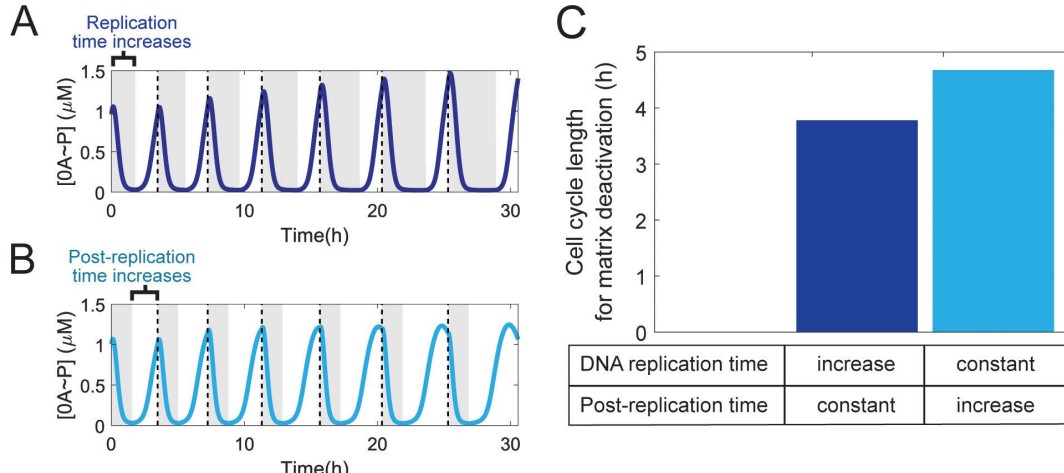

**Fig 5. DNA replication time is the cell-cycle stage that mostly contributes to biofilm matrix deactivation. (A)** 0A~P pulsing dynamics for increasing DNA replication period (grey regions) and constant post-replication period (white regions). Pulsing occurs once per cell-cycle. Dashed lines represent the end of a cell cycle. **(B)** The same analysis as (A) for constant DNA replication period and increasing post-replication period. **(C)** Cell cycle length for which biofilm matrix is deactivated when cell cycle time increases by solely increasing DNA replication period (dark blue bar) or the post-replication period (light blue bar).

Next, we tested the effects of these signals on matrix deactivation. We found that an increase in *OFF* time drives matrix deactivation, whereas an increase in *ON* time does not (Fig E, right panel, in S1 Text).

In the natural 0A~P pulsing signal, the equivalent of the *OFF* time period is the DNA replication period since the gene dosage ratio *kinA*:*0F* is approximately 1:2 during this period, resulting in low levels of 0A~P [17]. To test if the increase in the DNA replication period is the major regulator of biofilm matrix deactivation, we performed a similar test to the one for the simple model. Specifically, we simulated the phosphorelay network model under two different scenarios of cell-cycle slowdown (Methods Section 4.4). In the first one, the cell cycle and the DNA replication time increase by the same amount, and the post-replication period is kept constant (Fig 5A). In the second one, the cell cycle increases together with the post-replication period, but the DNA replication period is kept constant (Fig 5B). For each scenario, we tested the effects of the increase in the cell cycle on the deactivation of biofilm matrix production. We found that the biofilm matrix deactivates at shorter cell cycles for the network in which the DNA replication period increases, and the post-replication period is kept constant (dark blue bar in Fig 5C). However, unlike the simple model, we also observe deactivation for the 0A~P signal when only post-replication time increases (light blue bar in Fig 5C). This result suggests that post-replication time also plays a regulatory role in matrix deactivation, but its influence is smaller than that of the DNA replication period.

## 2.5 The growth rate influences biofilm deactivation through multiple mechanisms, with the contribution of the 0A~P pulsing period being the most significant

Previous research identified the growth effects on the biofilm network, namely the effects on gene dosage and protein dilution, as the primary regulators of biofilm matrix deactivation [34]. Here, we compared the impact of such effects to that of changes in 0A~P pulsatile signal dynamics in controlling biofilm matrix deactivation.

We began by simulating the biofilm matrix production network for five different conditions (Methods Section 4.4). Each condition differs in whether it includes the growth rate effects on gene dosage, protein dilution, and/or the 0A~P pulsatile signal. Specifically, starting with an initial condition where biofilm matrix production is active (set at a growth rate of 0.2 h$^{-1}$), we simulated the network, using 0A~P natural pulsing as input, to determine the minimum cell cycle length leading to biofilm matrix production deactivation (hereafter 'cell cycle deactivation threshold'). Specifically, we ran deterministic

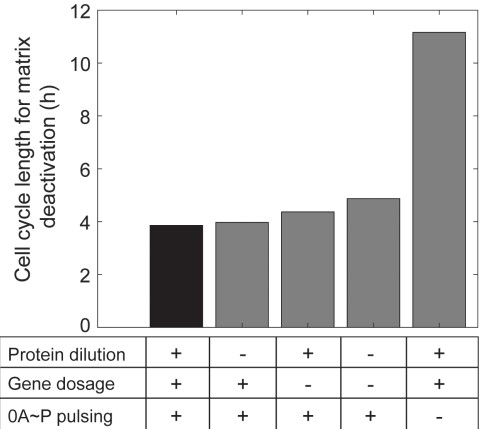

**Fig 6. 0A~P pulsing is the primary regulator of the minimum cell cycle length leading to biofilm matrix production deactivation.** Shown is the cell cycle length for which biofilm matrix is deactivated for five different networks differing in the presence (+) or absence (-) of growth rate effects on protein dilution, gene dosage, and/or 0A~P pulsing. The black bar indicates the result for the complete network with all growth rate effects present, assuming a growth rate equal to $0.2\,h^{-1}$.

simulations for increasing cell cycle lengths until reaching steady-state concentration of the model species (>100 h). Our results (black bar in Fig 6) show that the cell cycle deactivation threshold is 3.9 h. Next, for condition two, we examine how growth rate-mediated changes in protein dilution impact the cell cycle deactivation threshold. To this end, we simulated the network for effective protein degradation rates (Eq 16) without growth rate dependence (Eq 17). Under these conditions, matrix production deactivation occurs at a slightly longer cell cycle deactivation threshold, equal to 4 h. Similarly, in condition three, to examine the impact of gene dosage growth rate effects on the deactivation cell cycle threshold, we modified the average gene copy number (Eq 13) to be independent of growth rate. For this condition, matrix deactivation occurred at a cell cycle threshold equal to 4.4 h. For condition four, we simulated the network with the simultaneous removal of both these effects, which extended the deactivation cell cycle threshold to 4.8 hours (Fig 6). Thus, the effects of growth rate on gene dosage and protein dilution in the biofilm matrix production network change the deactivation threshold by less than 23%. In contrast, a much larger effect is observed for the final tested condition that assesses the impact of 0A~P pulsing dynamics, while maintaining all other growth rate effects. When we replaced the pulsatile 0A~P signal at a growth rate of 0.2 h⁻¹ with a constant 0A~P signal equivalent to the average value of the pulsatile signal, we observed an approximate 3-fold increase in the deactivation cell cycle threshold (Fig 6). We conclude that the 0A~P pulsatile signal is a key regulator of biofilm matrix deactivation, outweighing the previously identified effects of growth rate on gene dosage and protein dilution.

## 3. Discussion

Multiple single-cell phenotypes coexist in *B. subtilis* biofilms [6,37,38]. For example, spore-producing cells are often found near the aerial boundary of the biofilm [27]. It has been demonstrated that sporulating cells halt biofilm matrix production [39], likely because matrix production is an energy-intensive process requiring macromolecule synthesis, leading cells to reallocate energy towards sporulation [40–42]. To successfully achieve this outcome, the decision to initiate matrix production and later halt it and transition to sporulation must be tightly regulated. The gene regulatory networks controlling the initiation of matrix production have been extensively studied [3,6,42], highlighting the importance of precise transcriptional control in the synthesis of matrix components. In contrast, the mechanisms regulating the transition from a matrix production to a sporulating phenotype are less understood and subject of debate. To date, few transcriptional control

mechanisms have been proposed. First, high levels of 0A~P negatively regulate *sinI* expression, leading to repression of matrix production [39]. Second, changes in the gene dosage between *sinR* and *slrR* during sporulation are believed to repress matrix production [39]. However, recent studies showed that artificially induced high 0A~P levels can fail to repress matrix production genes [15], and instead, a slowdown in growth rate plays a critical role in altering the dosage of *sinR* and *slrR*, leading to the repression of matrix production [34]. Here, we describe another mechanism, based on the natural pulsatile dynamics of Spo0A~P, which has a stronger regulatory effect on the transition between matrix production and sporulation than all previously identified mechanisms. This mechanism not only enhances our understanding of matrix deactivation but also sheds light on the regulation of mutual exclusivity between biofilm and sporulation phenotypes.

Our results suggest that the period of 0A~P pulses, determined by the length of the cell cycle, may act as a key regulator of biofilm deactivation. Specifically, as the growth rate decreases (i.e., as the cell cycle lengthens), both the period and amplitude of 0A~P pulses increase (Fig 2A and 2B). However, for long cell cycles (i.e., low growth rates), maintaining biofilm activity requires an exceptionally high increase in 0A~P pulse amplitude (Fig 4C). Consequently, while the 0A~P pulse amplitude does increase, it fails to reach the threshold necessary to sustain biofilm matrix production. As a result, biofilm is deactivated. Nevertheless, the amplitude of the 0A~P signal has been shown to reach the threshold necessary to trigger sporulation [20]. Thus, these results suggest that the pulsing period may control biofilm deactivation whereas the pulsing amplitude triggers sporulation. This mechanism (Fig 7) provides further insight into how the mutual exclusivity of these two phenotypes is maintained.

Every cell cycle, the completion of DNA replication triggers a 0A~P pulse, which in turn serves as the activation signal for sporulation [17], thereby ensuring that sporulation is not triggered prior to DNA replication completion. Here we report

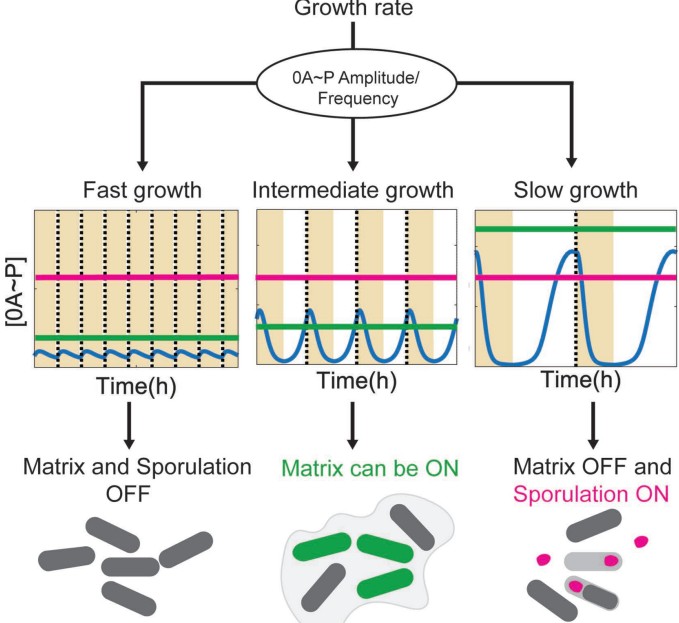

**Fig 7. 0A~P amplitude and frequency control cell fate decision between biofilm matrix production and sporulation. (Left panel)** In fast-growing cells (*i.e.,* under optimal conditions), biofilm matrix production and sporulation are not active. **(Middle panel)** At intermediate growth conditions (e.g., mild starvation), the amplitude of the 0A~P signal increases, its frequency decreases, and biofilm matrix production is activated stochastically in some cells (green line). The amplitude of the signal is not sufficient to trigger sporulation (pink line). **(Right panel)** Under slow growth conditions (e.g., prolonged starvation), the 0A~P amplitude increases, the frequency decreases, and the threshold to sustain matrix production (green line) increases (as shown in Fig 4C). The 0A~P signal fails to reach the threshold, and matrix production is deactivated. The 0A~P amplitude has been shown to reach the threshold necessary to trigger sporulation [20]. Yellow areas illustrate the DNA replication period. Dotted vertical lines represent the end of a cell cycle.

an additional feature controlled by the DNA replication period. Namely, beyond ensuring that sporulation occurs at the appropriate time, our results suggest that DNA replication may also be the cell-cycle stage that contributes most to the deactivation of biofilm matrix production (Fig 5C). This mechanism ensures that under rapid decrease of growth rate cells deactivate matrix production during the DNA replication period before peak 0A~P induces sporulation. As such, we propose that, during each cell cycle, cells reassess whether to continue biofilm production or cease it and initiate sporulation.

Interestingly, although the pulsing signal reaches higher levels than the corresponding average signal, we did not find evidence that 0A~P pulsing dynamics increase the activation of biofilm matrix production (Fig 3). Therefore, we conclude that activation is mostly controlled by the amplitude modulation of the 0A~P signal, whereas deactivation is controlled by the frequency modulation of the signal. Notably, frequency modulation has previously been reported for the alternative sigma factor, $\sigma^B$. Specifically, $\sigma^B$ regulates the *B. subtilis* stress response by controlling its target genes through sustained pulsing, where higher stress levels are encoded by increased pulse frequency. Altogether, this indicates that pulse modulation may be a widespread regulatory mechanism in bacteria [43].

While the models used in this study have been experimentally validated in prior research [20,34], we lack experimental validation of the results reported here. The major limitation to such validation is the absence of methods capable of precise dynamic control of transcription factors. Existing approaches, such as static gene perturbations (e.g., knockouts) [44] and basic dynamic control using chemical inducers [2], are not efficient for studying the fast natural temporal dynamics of transcription factors. Therefore, new approaches, for example, based on optogenetics or microfluidics [45] must be developed to enable precise, synthetic fine-tuning of transcription factor dynamics, to facilitate the *in vivo* study of how gene regulatory networks respond to such signals. For example, the light-sensitive two-component system CcaSR from Synechocystis PCC6803 was recently ported to *B. subtilis* to enable precise control of the transcription rate of a gene of interest using green and red light [46]. By controlling the transcription of Sad67, a constitutively active mutant of 0A [47] using CcaSR, synthetic oscillations of 0A could be produced in *B. subtilis* cells. The effect of oscillation parameters on cell fate decision-making could be examined using fluorescent protein-based transcriptional reporters (e.g., PTapA-GFP, pSpoIIQ-mCherry) [34]. Combination of this technique with a microfluidic device would allow examination of the effects of the pulsing dynamics at the single-cell level [48].

In the future, it would be interesting to investigate if pulsing 0A~P dynamics may play a role in committing to other cell fates besides sporulation and biofilm formation. For instance, the gradual accumulation of 0A~P has been shown to indirectly activate transcription of the master competence regulator ComK during early stationary phase, and to directly repress it at later stages, thereby limiting the window of competence to a subset of cells [49]. In addition, competence and sporulation have also been reported to be mutually exclusive cell-fates [50]. As such, one could investigate if pulsing of 0A~P regulates a cell's probability of becoming competent and the duration of the resulting competence window. Indeed, previous modeling work suggested that a repressilator-like network motif (consisting of Spo0A, Spo0E, and AbrB) generates pulses of ComK expression and periodic elevations in the probability of a transition to competence [51]. However, the impact of cell cycle-driven 0A~P oscillations on competence probability and competence window duration has not yet been investigated.

Moreover, pulsing dynamics of transcription factors have been shown to contribute to spatial pattern formation in developing biofilms. Specifically, [52] demonstrated that pulsing of $\sigma^B$ leads to a higher frequency of sporulating cells near the center of the biofilm, as opposed to the aerial boundary. It would be interesting to investigate whether oscillatory dynamics of 0A~P similarly influence the spatial patterning of population heterogeneity.

Overall, our results provide insight into how *B. subtilis* utilize dynamical 0A~P signaling to drive distinct stress-response phenotypes. Interestingly, mechanisms wherein TF activity patterns dictate distinct phenotypes have been reported for eukaryotes [53]. However, similar mechanisms remain largely understudied in bacteria. Given that the GRN controlling biofilm and sporulation are widely conserved in pathogenic bacterial species such as *B. anthracis* [54] and *B. cereus* [55], we expect our findings to have implications in understanding the survival mechanisms of these bacteria, which is critical

to understand their ability to cause infection. Ultimately, this study demonstrates the importance of dynamic transcription factor activity for bacterial cell fate decisions and limitations of the traditional steady-state approach.

## 4. Methods

### 4.1 Phosphorelay model

We extended a previous mathematical model of the 0A phosphorelay [20] to uncover the effects of 0A~P pulsing in the cell fate decision of *B. subtilis*. The reactions and parameters used in the model are listed in Table A in S1 Text, and the model differential equations are listed in Table B in S1 Text. The production of the phosphorelay proteins and phosphatases (KinA, Spo0F, Spo0B, Spo0A, Rap, and Spo0E) is modeled by reactions R1 to R6 in Table A in S1 Text. Post-translation interactions of phosphorylation/dephosphorylation are modeled by reactions R7 to R13. Specifically, after phosphorylation of the major sporulation kinase KinA (R7) the phosphoryl group is transferred to Spo0F (R8). Next, the phosphoryl group is transferred to Spo0B (R10) and finally to Spo0A (R11) [10,44]. Phosphorylated Spo0F and Spo0A are subject to negative regulation by phosphatases Rap (R12) and Spo0E (R13), respectively. Reaction (R9) accounts for the effect of inhibition of KinA by Spo0F.

The production initiation of phosphorelay proteins ($v_p^j$) is modeled with Hill-functions:

$$v_p^j = \left( v_b + v_{max} \cdot \frac{[0A \sim P]^m}{K^m + [0A \sim P]^m} \right) \cdot F(g)$$

(Eq 6)

Where, $v_b$ is the basal rate, $v_{max}$ is the maximum expression rate when promoter is fully induced. The variables $K$ and $m$ are the half-maximal binding constant and the Hill-exponent, respectively. The function $F(g)$ relates cell size with the growth rate ($g$). As in [20], we assume the existence of a delay $\left( \frac{1}{k_{del}} \right)$ between the change in protein production initiation rate ($v_p^j$) and the protein production rate at the current time ($v_p^c$).

$$\frac{dv_p^c}{dt} = k_{del} \cdot \left( v_p^j - v_p^c \right)$$

(Eq 7)

In addition, the rate of expression of all genes in the model was also assumed to be proportional to the gene copy number according to the following equation:

$$v = n \cdot v_p^c$$

(Eq 8)

As in [20], the original proximal gene *spo0F* was assumed to be replicated at the start of DNA replication whereas the terminus proximal gene (*kinA*) was assumed to be replicated by the end of DNA replication, as illustrated in Fig 1B. The DNA replication period ($\tau_{rep}$) was assumed to be a function of the growth rate ($g$) with the following expression [20]:

$$\tau_{rep} = 0.78 + \frac{0.15}{\mu}$$

(Eq 9)

With the total cell cycle period ($\tau_{cyc}$) being given by [20]:

$$\tau_{cyc} = \frac{ln(2)}{g}$$

(Eq 10)

Meanwhile, $F(g)$ follows the expression [20]:

$$F(g) = a \cdot e^{(b \cdot g)} + c \qquad \text{(Eq 11)}$$

The values of $a$, $b$, and $c$ were estimated from *in vivo* data in [20] and found to be $a = 3.5$, $b = -\ln(2)$, and $c = 3.7$.

### 4.2 Deterministic model of the biofilm matrix production network

To investigate the biofilm matrix production network assuming 0A~P pulsing dynamics we implemented a model based on a previous mathematical model of the biofilm matrix production network [34]. An illustration of the network is shown in Fig 1D. The model includes transcription, translation, and post-translation reactions of SinI, SinR, SlrR and TapA. The kinetic parameters and model reactions used can be found in Table C in S1 Text, while the corresponding differential equations are provided in Table D in S1 Text.

The transcription and translation of *sinI*, *sinR*, *slrR*, and *tapA* are modelled in reactions R1-R4 of Table C in S1 Text. Given that 0A~P induces the expression of *sinI*, the rate of production of SinI can be described according to the following equation:

$$v_{SinI} = \left( v_i^0 + v_i^{max} \cdot \frac{[0A \sim P]}{K_i + [0A \sim P]} \right) \cdot (n \cdot F(g)) \cdot \left( \frac{k_{tran}^i}{k_{deg}^m} \right) \qquad \text{(Eq 12)}$$

Where, transcription rate follows a Hill function with the basal transcription rate $v_i^0$ and $v_i^{max}$ the maximum transcription rate. The variable $K_i$ is the half-maximal constant, i.e., concentration of 0A~P at which the promoter is half-saturated. The variable $n$ corresponds to the average copy number and is estimated based on the following equation ([15]):

$$n = 2^{\left( 1 - \frac{\tau_{rep} \cdot p}{\tau_{cyc}} \right)} \qquad \text{(Eq 13)}$$

Where, $\tau_{rep}$ (DNA replication period) and $\tau_{cyc}$ (cell cycle length) are dependent on the growth rate as described by Eqs 9–10, respectively. The variable $p$ is the gene positioning relative to the *oriC*. As previously described, the function $F(g)$ relates cell size with growth rate and was set to follow the same expression as in [15]. The variables $k_{tran}^i$ and $k_{deg}^m$ are the translation and RNA degradation rate, respectively. We note that, both here and in Eqs 14 and 15, we chose to explicitly factor out the ratio $\left( \frac{k_{tran}^i}{k_{deg}^m} \right)$ from the production term, as this facilitates conversion to the stochastic biofilm production model, where transcription and translation are modeled as separate reactions to account for the effect of stochastic busts in transcription and translation.

We assume a constant rate of production for *sinR* (R2) since its expression is constitutive [56].

Meanwhile, given that the tetramer form of SinR (R6) acts as a repressor of *slrR* and *tapA*, similar to Eq 12, the rate of production of SlrR (R3) and TapA (R4) can be modelled according to the following equations:

$$v_{SlrR} = \left( v_l^0 + v_l^{max} \cdot \frac{K_l}{K_l + [R_T]} \right) \cdot (n \cdot F(g)) \cdot \left( \frac{k_{tran}^l}{k_{deg}^m} \right) \qquad \text{(Eq 14)}$$

$$v_{TapA} = \left( v_t^0 + v_t^{max} \cdot \frac{K_t}{K_t + [R_T]} \right) \cdot (n \cdot F(g)) \cdot \left( \frac{k_{tran}^t}{k_{deg}^m} \right) \qquad \text{(Eq 15)}$$

Where $v_{l/t}^0$ is the basal transcription rate, $v_{l/t}^{max}$ is the maximum transcription rate when the promoter is fully repressed by SinR tetramer form ($R_T$). The variable $K_{l/t}$ is the half-maximal binding constant. The subscripts $l$/$t$ refer to Eqs 14 and 15, respectively.

Moreover, SinR activity is regulated by SlrR and SinI due to complex formation. Namely, the SinI dimer ($I_d$) interacts with the SinR dimer ($R$) and forms a SinI-SinR heterodimer ($IR$) (reaction R7) [57]. On the other hand, SlrR dimer ($L$) associates with SinR dimer ($R$) and forms ($LR$) heterotetramer (R8). Notably, it is assumed that SinR [28] and SlrR are mostly dimeric [57].

The degradation rate of RNA ($k_{deg}^m$) was set to 8.3 h⁻¹ [58]. The degradation rate of all proteins ($k_{deg}^{pro}$) was set to 0.2 h⁻¹ [59], except for SlrR which was set to 0.8 h⁻¹, given that it is known to be an unstable protein [30]. In addition, since growth rate ($g$) affects protein dilution the effective protein degradation rate ($k_{eff}^{pro}$) was assumed to be given by:

$$k_{eff}^{pro} = k_{deg}^{pro} + g$$

(Eq 16)

As in [34], the relative transcription, translation and $K_i$ (Eq 12) rates were set to ensure the bifurcation diagram resulted in the transitions from matrix *OFF* state to matrix *ON* state at a realistic growth rate and Spo0A~P level (Fig 4).

### 4.3  Stochastic model of the biofilm matrix production network

Given that biofilm matrix activation occurs stochastically [34], the deterministic biofilm matrix production model (Tables C and D in S1 Text) was adapted into a stochastic framework to explore whether pulsatile dynamics of 0A~P influence the probability of matrix activation. The model reactions are shown in Table E in S1 Text. Unlike the deterministic model, we explicitly include the reactions that model gene ON–OFF switching dynamics driven by transcription factor binding/unbinding (R1 to R3 in Table E in S1 Text). Also, the stochastic model explicitly incorporates transcription (R4-R7) and translation reactions (R8-R11) for each species. This approach accounts for downstream effects of random bursts in transcription and translation, which are critical when evaluating matrix activation. Additionally, we also modeled the binding and unbinding dynamics of 0A~P to the promoter. This ensures that, under pulsatile 0A~P dynamics, the probability of promoter binding does not immediately follow the changes in 0A~P levels such that the time to reach the equilibrium probability of 0A~P binding to the promoter is accounted for. For the same reason, the binding and unbinding of the SinR tetramer to the *slrR* and *tapA* promoters is also explicitly modeled. Further, the transcription and translation rates for each mRNA were set to get a noise level in stochastic simulations to enable the stochastic activation and deactivation of biofilm matrix.

### 4.4  Model simulations

Deterministic model simulations (Figs 2, 4–6) were done using the *ode15s* solver of MATLAB R2023b. The stochastic model was implemented in MATLAB R2023b, and the stochastic simulations (Figs 3, D and E in S1 Text) were done using the *SimBiology* tool of MATLAB R2023b. The dynamics of the stochastic simulations follow the Stochastic Simulation Algorithm (also known as the Gillespie algorithm) [60,61]. The time length of each simulation was set to 40 h, found to be long enough to reach steady state. The results shown in Fig 3 are obtained from 2000 runs per condition, as this number suffices to obtain consistent results. The initial components at the start of simulations were set to be the final state of a 60 h preliminary stochastic simulation.

To generate Fig C in S1 Text, we extended the stochastic model to include other transcriptional noise sources. Specifically, we included in the model three new reactions that allow the promoter to enter a 'locked' state, in which transcription factor binding and transcription cannot occur (Table F in S1 Text). As previously, the dynamics of the stochastic simulations follow the Gillespie algorithm, and the maximum time of each simulation was set to 40 h. We performed 500 simulations per condition.

To generate Fig D in S1 Text, and evaluate the robustness of the results, we also developed a stochastic model that includes stochasticity in cell cycle dynamics. Specifically, we simulated cell cycles separately. At the beginning of each cell cycle, we sample the cell cycle duration $\tau_{cyc}$ (Eq 10) from a normal distribution with standard deviation equal to 0.1 [20]. After the cell cycle is sampled, the 0A~P level is calculated based on the growth rate, using the phosphorelay model. At

the end of each cell cycle, the cell volume is partitioned binomially [62] between the two daughter cells. One of the daughter cells is selected to continue the next cell cycle simulation. As previously, the dynamics of the stochastic simulations follow the Gillespie algorithm, and the maximum time of each simulation was set to 40 h. We performed 1000 runs per condition. We compare the results to those of a model that includes binomial partitioning of cell volume but lacks stochasticity in cell cycle dynamics (Fig D, black bars, in S1 Text).

In Fig 5, to investigate the role of DNA replication and post-replication period in deactivating biofilm matrix, Eq 9 was not used in the phosphorelay network to determine the DNA replication period and generate the corresponding 0A~P pulsing signal. Instead, model simulations were performed at an initial growth rate equal to 0.2 h$^{-1}$. This corresponds to a DNA replication period equal to 1.53 h and post-replication period equal to 1.94 h ($\tau_{cyc}$ equal to 3.47 h). Next, DNA replication period was increased by increments of 0.1 h while post-replication period was kept constant (Fig 5A). At each increment we generated the corresponding 0A~P pulsing signal and ran a deterministic simulation of the biofilm matrix production network until the cell-cycle period leading to matrix deactivation was found (dark blue bar in Fig 5C). To generate Fig 5B the opposite was done. Post-replication period was increased by increments of 0.1 h while the DNA replication period was kept constant.

To generate Fig 6, we simulated the biofilm matrix production network for five different conditions, each differing in the inclusion or exclusion of growth rate effects on gene dosage, protein dilution, and/or 0A~P pulsing. Specifically, assuming an initial condition for which biofilm matrix is active (set to be at a growth rate equal to 0.2 h$^{-1}$), we used the complete biofilm matrix production model with 0A~P pulsing as input to calculate the cell cycle length at which biofilm matrix production is deactivated. For this, we ran deterministic simulations for increasing cell cycle periods until reaching stead-state concentration of the model species (>100 h). Next, we determined what is the minimum period of the cell cycle leading to biofilm matrix deactivation.

To examine the impact of growth rate effects on protein dilution in relation to the deactivation cell cycle length, we modified the effective protein degradation rate to remove its dependence on growth rate. Consequently, Eq 16 was reformulated as:

$$k_{eff}^{pro} = k_{deg}^{pro} + g_0$$

(Eq 17)

Here, $g_0$ was set to 0.2 h$^{-1}$.

Similarly, to examine the impact of gene dosage growth rate effects on the deactivation cell cycle length, we modified the gene average copy number equation (Eq 13) to not be dependent on the growth rate. As such, $\tau_{rep}$ and $\tau_{cyc}$ are fixed at 1.53 h and 3.47 h, respectively. Finally, to assess the impact of 0A~P pulsing, we replaced the pulsatile 0A~P signal, at a growth rate of 0.2 h$^{-1}$, with a constant 0A~P signal equivalent to the average value of the pulsatile signal. Combined effects (e.g., deleting growth rate effects on protein dilution and gene dosage) were investigated by applying the above modifications simultaneously in the network.

### 4.5 Model fitting results

To estimate the biofilm matrix activation and deactivation rate ($k_{ON}$ and $k_{OFF}$, respectively), we simultaneously fit Eq 3 to 'Initially OFF' cells and Eq 4 to 'Initially ON' cells. We defined an objective function for optimization. This function minimizes the sum of the squared differences between the stochastic data (light blue and orange lines in Fig 3D) and predictions. Optimization was done using *fmincon* function of MATLAB R2023b.

In Fig 4D we use a phenomenological way to determine the threshold for which matrix production is deactivated ($M_{OFF}$), i.e., the average 0A~P concentration for which the system shifts from an active to a deactivated matrix production state). Specifically, we fit a power-law approximation (Eq 18) using the non-linear least square method to fit the data points for which 0A~P pulsing period and cell cycle ($\tau_{cyc}$) are the same (blue circles in Fig 4D).

$$M_{OFF}[\mu M] = a \cdot (\tau_{cyc}[h])^b + c \qquad \text{(Eq 18)}$$

The fitting results are $a = 0.06$, $b = 2.59$, and $c = 0.14$. The fitting $R^2$ value is 0.99.

## Supporting information

**S1 Text. Supplementary information.** Supplementary tables A-F and supplementary figures A-E.
(PDF)

## Author contributions

**Conceptualization:** Cristina S.D. Palma, Daniel J. Haller, Jeffrey J. Tabor, Oleg A Igoshin.

**Data curation:** Cristina S.D. Palma.

**Formal analysis:** Cristina S.D. Palma.

**Funding acquisition:** Jeffrey J. Tabor, Oleg A Igoshin.

**Investigation:** Cristina S.D. Palma, Daniel J. Haller.

**Methodology:** Cristina S.D. Palma.

**Software:** Cristina S.D. Palma.

**Supervision:** Jeffrey J. Tabor, Oleg A Igoshin.

**Visualization:** Cristina S.D. Palma, Daniel J. Haller.

**Writing – original draft:** Cristina S.D. Palma, Daniel J. Haller.

**Writing – review & editing:** Cristina S.D. Palma, Daniel J. Haller, Jeffrey J. Tabor, Oleg A Igoshin.

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
