## [Decision Letter · Decision Letter 0]

Changes in Spo0A~P pulsing frequency control biofilm matrix deactivation

PLOS Computational Biology

Dear Dr. Igoshin,

Thank you for submitting your manuscript to PLOS Computational Biology. After careful consideration, we feel that it has merit but does not fully meet PLOS Computational Biology's publication criteria as it currently stands. Therefore, we invite you to submit a revised version of the manuscript that addresses the points raised during the review process.

Please submit your revised manuscript within 60 days Jul 07 2025 11:59PM. If you will need more time than this to complete your revisions, please reply to this message or contact the journal office at ploscompbiol@plos.org. Please include the following items when submitting your revised manuscript:

We look forward to receiving your revised manuscript.

Kind regards,

Nic Vega, Ph.D.

Academic Editor

PLOS Computational Biology

Stacey Finley, Ph.D.

Section Editor

PLOS Computational Biology

**Additional Editor Comments :**

The reviewers were overall positive about the manuscript, suggesting additional clarification on some of the model assumptions and results, as well as a small number of additional simulations. However, some of the requested additional work may be achievable with text edits (e.g. to clarify the rationale behind certain simplifying assumptions).

**Journal Requirements:**

At this stage, the following Authors/Authors require contributions: C.S.D. Palma, Daniel J. Haller, Jeff J. Tabor, and Oleg A Igoshin. Please ensure that the full contributions of each author are acknowledged in the "Add/Edit/Remove Authors" section of our submission form.

4) We notice that your supplementary Figures, and Tables are included in the manuscript file. Please remove them and upload them with the file type 'Supporting Information'. Please ensure that each Supporting Information file has a legend listed in the manuscript after the references list.

Potential Copyright Issues:

i) We note that Figure 1 is created through BioRender. Please confirm that you hold a Premium account and provide a pdf copy of the CC BY 4.0 Licence as provided by BioRender. For instructions on how to generate a CC BY 4.0 license for your figure, please see the guidelines here: https://help.biorender.com/hc/en-gb/articles/21282341238045-Publishing-in-open-access-resources. 

If you are using the free assets from BioRender, we are unable to publish these images as they are licenced under a stricter licence than CC BY 4.0. In this case we ask you to remove the BioRender images and replace them with open source alternatives.

See these open source resources you may use to replace images / clip-art:

- https://bioart.niaid.nih.gov/ 

- https://bioicons.com/

- https://healthicons.org/ 

- https://scidraw.io/

- https://reactome.org/icon-lib

- https://www.phylopic.org/images

7) Please ensure that the funders and grant numbers match between the Financial Disclosure field and the Funding Information tab in your submission form. Note that the funders must be provided in the same order in both places as well. Currently, the order of the funders is different in both places.

**Reviewers' comments:**

Reviewer's Responses to Questions

Reviewer #1: In this paper, a mathematical model is developed and analyzed in order to investigate the role of the pulsing frequency of the Spo0A-P transcription factor as a regulator of biofilm production by the B. subtilis bacterium. More precisely, the Spo0A-P transcription factor has already been shown to exhibit an oscillatory dynamics, with oscillation frequency that depends on the cell growth rate. Moreover, the cell growth rate has already been shown to be a controller of differentiation into biofilm matrix-producing cells or sporulating cells. The hypothesis investigated in this paper is whether the Spo0A-P oscillation frequency could be the growth rate-mediated mechanism to control differentiation.

The authors develop a composite mathematical model that consists of two deterministic modules (phosphorelay network and biofilm matrix production network) and a stochastic module modeling aspects of biofilm matrix production that can be influenced by stochastic noise (e.g. transcription and translation of relevant genes). The model is then analyzed by performing simulations in MATLAB.

Simulation results show that the oscillatory dynamics of Spo0A-P plays a control role for biofilm production, in particular in the phase of deactivation of such an activity: higher oscillatory periods (with the same mean) turn out to favor biofilm production deactivation.

The paper is interesting, well-written and clear. Provided that Spo0A-P has already been shown to have an oscillatory dynamics influenced by the cell growth rate, the hypothesis that its pulsing frequency can play a control role in biofilm matrix production seems reasonable and is supported in a convincing way by the in-silico analysis performed in this paper. Both modeling and analysis are performed in a rigorous way.

The results obtained in this paper should now undergo a validation phase through in vitro experiments. However, the authors clearly discuss the difficulties of conceiving in vitro experiments for testing the role of oscillatory behaviors. Although these difficulties in validating the results could be seen as a weakness, I think that they actually further motivate the in-silico study that (if done in a rigorous way) can provide a significant support for the proposed hypothesis.

Prior to publication, the paper should be improved in a few aspects. Firstly, in the first part of the analysis (Fig.2) the authors compare the effects of a pulsing dynamics with those of non-pulsing average level. Although this first comparison allows the reader to understand the hypothesis the authors aim to investigate, by reading this part I thought that a comparison with other OA-P levels (e.g., min and max) should be added here. The role of OA-P concentration levels is investigated later in the paper, but here I had the impression that the comparison was not complete.

Another aspect that I did not find completely convincing is the imposition of a single k_on in the fitting of curves in Fig. 3D. According to what written in the paper, this is motivated only by the empirical observation of the initial slope of the curves in the figure. It seems to me a rather weak motivation, so I suggest the authors to repeat the curve fitting without this constraint and let such an equivalence to emerge from the data fitting.

Minor issues:

- line 93: clarify that data in Figure 1C are simulation results. Since the mathematical model has not been introduced at this point, I initially thought these to be data from wet experiments

- Figure 1C: the graph on the right is not clear. Too dense and difficult to read

- line 141: the wording "incorrect conclusions" is too strong. Although the analysis in this paper is convincing, the control role of 0A-P oscillations is still a hypothesis.

Reviewer #2: In this interesting study Palma et al. investigates how the pulsing dynamics of Spo0A regulate the deactivation of biofilm matrix production in Bacillus subtilis. By integrating deterministic and stochastic models of the Spo0A phosphorelay and the biofilm regulatory network, the authors propose that the frequency of Spo0A pulses drives matrix production deactivation under starvation. Simulations show that as growth rate slows, changes in cell cycle timing alter Spo0AP pulsing dynamics, which in turn bias cells toward sporulation by shutting off matrix production. Overall, the study presents a convincing case for their proposed mechanism.

The manuscript is well written, and the modeling is well executed and clearly described. Although the conclusions are not validated experimentally, the modeling is valid and interesting in its own right. However, the manuscript could benefit from minor revision.

Major Comment

Further investigation of robustness of predictions - The conclusion that pulsing frequency controls biofilm deactivation is plausible, but given the lack of experimental validation it would be great to have further tests of the robustness of these conclusions in terms of the model assumptions. Could the authors show further that the results do not depend on specific modeling assumptions?

e.g Could the authors examine whether key findings are affected if the 200 molecule TapA threshold for defining ON/OFF states is varied? For the stochastic modeling, what would happen if transcriptional bursting is incorporated or other methods of simulating the stochastic model are used? What would happen if cell volume effects or stochasticity in cell cycle dynamics are incorporated in the stochastic model? Discussion of these points would help demonstrate further that the central conclusions are not due to particular model features.

Minor points

Could the authors provide more details about the stochastic model and how the equations were simulated? I could not see details about the algorithm used (e.g Gillespie?) and the number of stochastic simulations carried out, but might have missed it.

The authors mention microfluidics and optogenetics will in future allow tests of the model, but can the authors spell out a bit more how these techniques can allow this - ie how the tests would be carried out?

Some claims could be tempered a little - (e.g that frequency controls deactivation), given the lack of experimental validation, unless it is the authors view that this is the only plausible mechanism?

The authors could consider referencing previous work examining competition between sporulation and other alternative cell fates in B. subtilis -e.g. (doi: 10.1371/journal.pgen.1002586, https://doi.org/10.1038/msb.2011.88, https://doi.org/10.1038/s41467-020-14431-9).

Line – 630 - spelling mistake – gradualincrease

Reviewer #3: The study by Palma et al uses mathematical models to propose an alternative mechanism of cell fate decision in Bacillus subtilis. Cell differentiation involves the phosphorylated transcription factor SpoA~P, which is known to activate two differentiation pathways that are mutually exclusive at the single-cell level: biofilm formation or sporulation. In previous studies, the authors showed that the decision is controlled by the effects of growth rate on gene dosage and protein dilution in the biofilm matrix production network. This study extends the previous findings by showing that growth-rate dependent changes of the pulsing period of SpoA~P also controls matrix deactivation and this appears to be the primary regulator.

The mathematical models of the phosphorelay and biofilm production networks have been published previously. They are used in a different way in this study. Combining the deterministic models of these two networks allows the role of SpoA~P pulsing on matrix deactivation to be demonstrated. Simulations with the stochastic version of the biofilm production network model show that the pulsing dynamics specifically affect the deactivation rate, but not the activation rate. More precisely, the period of SpoA~P oscillations controls the biofilm deactivation threshold, as shown by the deterministic model of the biofilm production network coupled to SpoA~P oscillatory signals. Additional simulations modifying the nature of the pulsing signal show that within the cell cycle, the DNA replication phase is most responsible for controlling matrix deactivation. Finally, the relative contributions of the different mechanisms influencing biofilm deactivation (gene dosage, protein dilution and SpoA~P pulsing) are assessed using the deterministic model of biofilm production.

The use of the different models to dissect the role of SpoA~P pulsing dynamics on matrix deactivation is elegant. The results are very convincing and advance knowledge on cell fate decision in B. subtilis. The paper is in addition clear and well written. MATLAB scripts are provided to reproduce the simulations and results shown in the figures of the paper.

A few elements need to be clarified:

- There is a rescaling of the transcription rate by the degradation rate (k_tran/k_deg) in Equations 14, 16-17: what is the rationale behind this formulation in the biofilm production model, which is not present in the phosphorelay model?

- The degradation terms in the MATLAB file for the biofilm model (SinIR.m) are proportional to the degradation constant: is the rescaling k_tran/k_deg implemented in a different manner in the code? If yes, why?

- Without a very good knowledge of the authors’ previous work, it is difficult to understand all the work carried out in the paper: it would help the reader if the full models (stochastic and deterministic) were given in the supplementary text even if they have been published in large parts in previous studies (I had to go through the Matlab files to understand the model equations).

Minor comments:

- Discrepancy between Equation 10 in the main text and Tables S1, S2: the whole term for the gene expression rate should be multiplied by [n.F(g)]. The MATLAB code is in agreement with the expression in Equation 10.

- Equations R1 to R13 in Table S1 and R5 to R8 in Table S2: The kinetic rate laws should also be provided, even if these are simple (as explained above, the complete list of model equations & rate laws would be easier to understand)

- Error in the biochemical reaction for reaction R8 in Table S1: we should read KinA + OFp rather than OA + OE

- Is reaction R9 needed as it is already present in reaction R8?

- Concerning the Matlab code:

* README text for Fig1C: remove ‘_02’ in call to Matlab function

* Redundant README files in Fig2 folder

* Script ‘Bifurcation_amplitude_Average.m’: problem with the limit of the vertical axis

* Missing README file in Fig 3 folder

**Have the authors made all data and (if applicable) computational code underlying the findings in their manuscript fully available?**

Reviewer #1: Yes

Reviewer #2: Yes

Reviewer #3: Yes

PLOS authors have the option to publish the peer review history of their article (what does this mean? ). If published, this will include your full peer review and any attached files.

**Do you want your identity to be public for this peer review?** For information about this choice, including consent withdrawal, please see our Privacy Policy .

Reviewer #1: **Yes: ** Paolo Milazzo

Reviewer #2: No

Reviewer #3: No

**Figure resubmission:**
---

## [Editor Report · Decision Letter 1]

Dear Prof. Igoshin,

We are pleased to inform you that your manuscript 'Changes in Spo0A~P pulsing frequency control biofilm matrix deactivation' has been provisionally accepted for publication in PLOS Computational Biology.

Best regards,

Nic Vega, Ph.D.

Academic Editor

PLOS Computational Biology

Stacey Finley, Ph.D.

Section Editor

PLOS Computational Biology

---

## [Editor Report · Acceptance letter]

PCOMPBIOL-D-25-00289R1

Changes in Spo0A~P pulsing frequency control biofilm matrix deactivation

Dear Dr Igoshin,

I am pleased to inform you that your manuscript has been formally accepted for publication in PLOS Computational Biology. Your manuscript is now with our production department and you will be notified of the publication date in due course.

With kind regards,

Anita Estes
